# A DFT−Based Mechanism Analysis of the Cyclodextrin Inclusion on the Radical Scavenging Activity of Apigenin

**DOI:** 10.3390/antiox12112018

**Published:** 2023-11-20

**Authors:** Xiaoping Zheng, Yapeng Du, Yu Chai, Yanzhen Zheng

**Affiliations:** College of Ocean Food and Biological Engineering, Jimei University, Xiamen 361021, China; grfafu126@126.com (X.Z.); dfcfafu126@126.com (Y.D.); fzmfafu126@126.com (Y.C.)

**Keywords:** apigenin, flavonoid, antioxidative activity, density functional theory, cyclodextrin inclusion

## Abstract

Natural flavonoids are renowned for their exceptional antioxidant properties, but their limited water solubility hampers their bioavailability. One approach to enhancing their water solubility and antioxidant activity involves the use of cyclodextrin (CD) inclusion. This study investigated the impact of CD inclusion on the three primary radical scavenging mechanisms associated with flavonoid antioxidant activity, utilizing apigenin as a representative flavonoid and employing density functional theory (DFT) calculations. Initially, the optimized geometries of CD−apigenin inclusion complexes were analyzed, revealing the formation of hydrogen bonds between CD and apigenin. In less polar environments, the inclusion complex strengthened the bond dissociation enthalpies of hydroxyl groups, thereby reducing antioxidant activity. Conversely, in polar environments, the inclusion complex had the opposite effect by lowering proton affinity. These findings align with experimental results demonstrating that CD inclusion complexation enhances flavonoid antioxidant activity in aqueous ethanol solutions.

## 1. Introduction

Free radicals, comprising reactive oxygen species characterized by unpaired electrons, are prevalent in the human body and can inflict substantial harm on biomacromolecules like lipids, proteins, and DNA [1,2]. Such damage is associated with a myriad of diseases, spanning cardiovascular disorders, cancer, atherosclerosis, and neurodegenerative conditions [3,4,5].

Antioxidants play a crucial role in capturing free radicals and mitigating their damage. Among these antioxidants, flavonoids constitute a widespread category of polyphenolic compounds present in everyday foods like fruits, vegetables, cereals, and bee products. Flavonoids are effective scavengers of various free radicals, including hydroxyl, peroxyl, and lipid peroxyl radicals [6,7]. Their remarkable antioxidative capacities, characterized by powerful chain-breaking actions and the ability to neutralize free radicals, shield biomacromolecules from the detrimental effects of oxidative stress [8]. Beyond their role as antioxidants, flavonoids offer a range of health benefits, including anti-inflammatory, antiallergic, antiviral, and anticancer properties [9,10,11].

Like other phenolic compounds, flavonoids possess relatively low water solubility and are susceptible to in vivo enzymatic degradation [12], factors that severely limit their bioavailability and practical applications. Given the significance of flavonoids, the development of efficient water-soluble methods has become increasingly crucial. Among various techniques, the formation of inclusion complexes with other water-soluble components stands out as an attractive and effective approach. Cyclodextrins (CDs) are cyclic oligosaccharides comprising six (*α*−CD), seven (*β*−CD), eight (*γ*−CD), or more glucopyranose units [13]. They feature a truncated cone structure with a hydrophobic cavity and hydrophilic exterior [13]. The cavities of the *α*−CD, *β*−CD, and *γ*−CD are 0.47–0.53 nm, 0.60–0.65 nm, and 0.75–0.83 nm, respectively [14]. This unique structure renders CDs ideal for solubilizing nonpolar or weakly polar solutes in their hydrophobic cavity while enabling them and the solutes to become water soluble due to the hydrophilic exterior [13,15]. Research has demonstrated that the water solubility and bioavailability of flavonoids can be significantly enhanced by forming inclusion complexes with CDs [16,17]. Additionally, improvements in antioxidative capacities and enhanced stabilities of the included flavonoids against oxidation by free radicals have also been observed [18,19].

Antioxidants function to protect against free radicals through three primary mechanisms: Hydrogen Atom Transfer (HAT), Single Electron Transfer−Proton Transfer (SET−PT), and Sequential Proton Loss Electron Transfer (SPLET) [20,21,22]. While experimental data have shown improved antioxidative activity in CD inclusion complexation with flavonoids, the precise mechanism underlying the outstanding antioxidative capacity of the CD−flavonoid complex remains unclear. Furthermore, the impact of inclusion on antioxidative activity has not been fully elucidated. Therefore, in-depth discussions and investigations are warranted to shed light on these aspects.

In this study, we conducted comparative analyses to elucidate the antioxidative activity difference of flavonoids before and after CD inclusion complexation through the mechanisms of HAT, SET−PT, and SPLET. Apigenin (Figure 1), among the most abundant and extensively studied flavonoids, is found in significant quantities in various natural foods such as parsley, celery, thyme, onions, honey, and bee pollen. In this research, apigenin was chosen as the representative flavonoid for detailed investigation. In recent years, numerous computational studies have focused on unraveling the antioxidative activities of phenolic compounds. Density Functional Theory (DFT) calculations have emerged as the most suitable computational method for assessing the antioxidative properties of flavonoids due to their high precision, cost-effectiveness, and efficiency in handling calculations for medium-sized and large molecules. DFT calculations have been successfully employed in studying the antioxidative properties of various phenolic compounds [20,21,22,23,24,25,26,27,28,29,30] and served as the primary research tool in this study. Considering the influence of solvent effects on the thermodynamically favored mechanisms, we considered seven solvent phases in our analysis. These solvent phases, with varying polarities, comprehensively encompass a range of possible environments. This approach allowed us to make informative comparisons based on the results obtained from DFT calculations.

## 2. Computational Details

All computations were executed employing the Gaussian 16 package [31]. The structural information for *α*−CD and *β*−CD was sourced from the Cambridge Crystallographic Data Centre. The positions of hydrogen atoms within *α*−CD and *β*−CD were optimized using the B3LYP/6-31G* method [32,33]. For neutral molecules, radicals, cation radicals, and anions, all equilibrium geometries and vibrational frequencies were optimized using the M062X/6-311+G** method [33,34], ensuring the absence of imaginary vibrational frequencies to confirm the stability of minima geometries. Additionally, to obtain precise interaction energies between CD and apigenin, basis set superposition error (BSSE) correction was applied [35]. Single-point energies, as well as distributions and energies of the frontier orbitals of apigenin, *α*−CD−apigenin, and *β*−CD−apigenin complexes, were computed using the M062X/6-311+G** method. The solvent effect was incorporated utilizing the SMD continuum solvent model [36].

### 2.1. Atoms in Molecules Analysis

The atoms in molecules (AIM) theory serves as a valuable tool for studying hydrogen bond properties [37]. This theoretical framework is rooted in topological theory and describes chemical bonds based on their topological features. Within AIM theory, a chemical bond is characterized by a bond critical point (BCP) located along the bond path between the hydrogen donor and the hydrogen bond acceptor. Various topological parameters, including electron density (*ρ*_BCP_), the Laplacian of electron density (∇^2^*ρ*_BCP_), Lagrangian kinetic energy (*G*_BCP_), potential electron density (*V*_BCP_), and energy density (*H*_BCP_) at the BCP, are employed to investigate the properties of the hydrogen bond. In this study, the identification of BCPs in hydrogen bonds and detailed topological analysis were conducted using the Multiwfn 3.5 suite [38].

### 2.2. Antioxidative Mechanisms

The details of the three mechanisms are as follows [20,21,22]:(1)HAT mechanism:
R• + ArOH → RH + ArO•(1)

(2)SET−PT mechanism:

R• + ArOH → R^−^ + ArOH•^+^(2)

R^−^ + ArOH•^+^ → RH + ArO•(3)

(3)SPLET mechanism:

ArOH → ArO^−^ + H^+^(4)

ArO^−^ + R• → ArO• + R^−^(5)

In Equations (1)–(5), R•, ArOH, ArO•, ArOH•^+^, and ArO^−^ represent for free radical, neutral flavonoid, flavonoid radical, flavonoid cation radical, and flavonoid anion, respectively. The reaction enthalpies related to Equations (1)–(5) are usually denoted as follows:

BDE: bond dissociation enthalpy (BDE) related to (Equation (1)). The calculated equation for BDE is:BDE = H(ArO•) + H(H•) − H(ArOH)(6)

IP: ionization potential (IP) related to (Equation (2)). The calculated equation for IP is:IP = H(ArOH•^+^) + H(e^−^) − H(ArOH)(7)

PDE: proton dissociation enthalpy (PDE) related to (Equation (3)). The calculated equation for PDE is:PDE = H(ArO•) + H(H^+^) − H(ArOH•^+^)(8)

PA: proton affinity (PA) of phenoxide anion (Equation (4)). The calculated equation for PA is:PA = H(ArO^−^) + H(H^+^) − H(ArOH)(9)

ETE: electron transfer enthalpy (ETE) related to (Equation (5)). The calculated equation for ETE is:ETE = H(ArO•) + H(e^−^) − H(ArO^−^)(10)

The calculated solvent phases enthalpies for H^+^, e^−^ and H• were obtained from the references [39,40,41].

## 3. Results

### 3.1. Conformational Properties

Before proceeding with the analysis of various molecular descriptors, we conducted an exploration of the conformational properties of apigenin and the CD−apigenin complex. The most stable structures of the *α*−CD−apigenin complex and the *β*−CD−apigenin complex are depicted in Figure 1A,B, respectively, with the respective interaction energies indicated below the structures. Notably, the interaction between *β*−CD and apigenin is observed to be stronger than that between *α*−CD and apigenin. Hydrogen bonding is an important interaction in the inclusion process. We employed the sum of van der Waals atomic radii for hydrogen and oxygen (2.5 Å) to determine the presence of hydrogen bonds in the CD−apigenin complex [42]. For the purpose of this study, our subsequent analyses primarily focused on apigenin. To provide a clearer visualization, Figure 1C,D display only apigenin and the interacting atom of CD.

The distances between the interacting hydrogen atom and oxygen atom within the complexes have been carefully labeled. These distances, which are less than 2.5 Å, unequivocally confirm the presence of intermolecular hydrogen bonds between apigenin and CD. To delve deeper into the nature of these hydrogen bonds within the complex, we examined the topological parameters utilizing AIM theory. The values of *ρ*_BCP_, ∇^2^*ρ*_BCP_, *G*_BCP_, *V*_BCP_, and *H*_BCP_ at the BCP for the hydrogen bonds in both apigenin and the CD-apigenin complex are presented in Table 1.

The analysis in Table 1 reveals that the *ρ*_BCP_ values, ranging from 0.010 to 0.054 a.u., fall within the established range for hydrogen bonds [43]. *ρ*_BCP_ serves as a numerical indicator of hydrogen bond strength, with higher values indicating stronger bonds. Consequently, the intramolecular hydrogen bond (O5−H5⋯O4) in apigenin weakens in the CD−apigenin complex. Notably, the O4 atom emerges as the predominant interaction site, characterized by the highest *ρ*_BCP_ value of O4⋯H−O among all hydrogen bond interactions.

The range of ∇^2^*ρ*_BCP_ values observed in this study (ranging from 0.035 to 0.171 a.u.) aligns with established values for hydrogen bonds [43], confirming the presence of hydrogen bond interactions in both apigenin and the CD−apigenin complex. The sign of ∇^2^*ρ*_BCP_ serves as an indicator of the hydrogen bond type, with negative values denoting shared-type interactions and positive values denoting closed-shell interactions [38]. Consequently, the positive ∇^2^*ρ*_BCP_ value reported in Table 1 for the CD-apigenin complex confirms the prevalence of closed-shell type hydrogen bond interactions in this system.

The strength of a hydrogen bond can be classified into three types [44]:(1)Weak hydrogen bond: ∇^2^*ρ*_BCP_ > 0 and *G*_BCP_ + *V*_BCP_ > 0;(2)Medium hydrogen bond: ∇^2^*ρ*_BCP_ > 0 and *G*_BCP_ + *V*_BCP_ < 0;(3)Strong hydrogen bond: ∇^2^*ρ*_BCP_ < 0 and *G*_BCP_ + *V*_BCP_ > 0.

Table 1 reveals exclusively positive values for ∇^2^*ρ*_BCP_. The O5−H5⋯O4 intermolecular hydrogen bond in apigenin and the bonds between *β*−CD and apigenin exhibit negative *G*_BCP_ + *V*_BCP_ values, indicating their moderate strength. In contrast, the hydrogen bonds involving *α*−CD and apigenin display positive *G*_BCP_+*V*_BCP_ values, signifying their relatively weaker strengths.

The sign of *H*_BCP_ can be used to distinguish whether a hydrogen bond is of a covalent or electrostatic nature [41]. The negative sign of *H*_BCP_ implies the hydrogen bond possessing a dominant character of covalent interactions. While the positive value indicates the electrostatic dominant nature of a hydrogen bond [45]. From Table 1, it is evident that the *H*_BCP_ is negative for the hydrogen bonds in apigenin and *β*−CD−apigenin complex, whereas the *H*_BCP_ is positive for the hydrogen bonds in the *α*−CD−apigenin complex. Therefore, the hydrogen bonds in apigenin and *β*−CD−apigenin complex possess a dominant character of the covalent interactions, whereas those in *α*−CD−apigenin complex are electrostatic dominant.

### 3.2. Frontier Orbitals Analysis

The primary orbitals involved in chemical reactions are the highest occupied molecular orbital (HOMO) and the lowest unoccupied molecular orbital (LUMO). Figure 2 illustrates the distribution and energies of these frontier orbitals in the gas phase for apigenin. The visualization in Figure 2 indicates that both the HOMO and LUMO orbitals are predominantly spread across the molecule.

The energy of the HOMO serves as a crucial parameter in representing the antioxidative activity of phenolic compounds, capturing both hydrogen atom/proton donation and electron transfer processes employed to trap free radicals. Phenolic compounds with higher HOMO energy exhibit stronger antioxidative activity. Figure 2 illustrates that the apigenin monomer possesses the highest HOMO energy, succeeded by *β*−CD−apigenin and α−CD−apigenin. This finding unequivocally establishes apigenin monomer as having the most robust electron-donating capability. The complexation with cyclodextrins, particularly α−CD, diminishes the electron-donating potential of apigenin.

According to frontier orbital theory, the flavonoid’s most active redox site is identified by a high density of the HOMO. In our study, the HOMO orbitals are localized in three hydroxyl positions within apigenin. Consequently, all hydroxyl groups in apigenin are susceptible to attack by electrophilic agents like free radicals [24,26,29,30].

### 3.3. HAT Mechanism

The HAT mechanism is defined by the BDE. A lower BDE indicates an easier reaction and, consequently, a higher antioxidative activity. The calculated BDE values for the hydroxyl groups in apigenin are compiled in Table 2.

Table 2 shows that the BDE of 4′−OH is smaller than that of the other hydroxyl groups, indicating that the abstraction of the hydrogen atom from 4′−OH is easier compared to other hydroxyl groups. Interestingly, the most potent antioxidative hydroxyl group in the CD−apigenin complex remains consistent with apigenin monomer across different phases. In antioxidants with multiple phenolic hydroxyl groups, the free radical scavenging activity is determined by the one with the lowest O−H BDE in the HAT mechanism. Evaluating the lowest O−H BDEs, the hydrogen-donating ability of apigenin follows this sequence: apigenin monomer > *α*−CD−apigenin > *β*−CD−apigenin. Consequently, apigenin appears to be the most effective scavenger following the HAT mechanism, and the complexation with cyclodextrin diminishes its antioxidative capacity.

In the case of 4′−OH and 7−OH, the BDE increases upon complexation with CD. Consequently, the free radical scavenging capacity of 4′−OH and 7−OH in the CD−apigenin complex becomes weaker compared to apigenin alone. Moreover, in the *α*−CD−apigenin complex, the homolytic cleavage of 7−OH requires breaking an intermolecular hydrogen bond (as illustrated in Figure 1C). Therefore, the hydrogen atom abstraction from 7−OH in the α−CD−apigenin complex is more challenging than in the *β*−CD−apigenin complex and apigenin. Similarly, the same difficulty arises for 4′−OH in the *β*−CD−apigenin complex (as depicted in Figure 1D).

In contrast to 4′−OH and 7−OH, the BDE of 5−OH decreases after complexation with CD. Consequently, the antioxidative activity of 5−OH in the CD-apigenin complex is more potent than in apigenin alone. This enhancement can be linked to the weakening of the O5−H5···O4 intramolecular hydrogen bond. A weaker O5−H5···O4 hydrogen bond correlates with a stronger antioxidative activity of 5−OH.

Based on the preceding analysis, it can be inferred that the hydrogen bonds in the CD-apigenin complex significantly impact the antioxidative activity of apigenin in the context of the HAT mechanism.

### 3.4. SET-PT Mechanism

#### 3.4.1. Ionization Potential

In the SET-PT mechanism, electron donation initiates the process, and IP stands as a vital parameter indicating the electron donation capability. The calculated IP values for apigenin in the examined phases are detailed in Table 3.

Table 3 shows an increase in the IP value for CD−apigenin, with *α*−CD−apigenin displaying the highest IP value across all studied phases, consistent with the findings in Section 3.2. This consistency may be attributed to the instability of the cation radical of apigenin, leading to more localized delocalization and conjugation of its π−electrons within the CD−apigenin complex, particularly in the *α*−CD−apigenin complex. IPs of apigenin and the *β*−CD−apigenin are practically the same due to fact that the differences are lower than 5 kJ/mol and complexation may not affect the electron-donating ability. Consequently, the complexation with *α*−CD diminishes the electron-donating ability of apigenin.

#### 3.4.2. Proton Dissociation Enthalpy

In the SET-PT mechanism, the second step involves proton donation from the cation radical, a process characterized by PDE. A lower PDE value signifies a stronger proton-donating ability of the cation radical. The calculated PDE values for various hydroxyl groups in apigenin are outlined in Table 3.

In Table 3, PDEs obey the trends found for BDEs. It is evident that the PDE of 4′−OH is smaller than that of the other hydroxyl groups, confirming that proton donation from 4′−OH in the apigenin cation radical is easier than from other hydroxyl groups. Additionally, the most probable proton-donating group in the apigenin cation radical remains consistent in different phases despite CD inclusion complexation. Based on the calculated lowest O−H PDEs, the sequence of proton-donating ability in the apigenin cation radical is as follows: *α*−CD−apigenin > apigenin monomer > *β*−CD−apigenin. Therefore, the apigenin cation radical in *α*−CD−apigenin exhibits the strongest proton-donating ability.

The PDEs of 5−OH in the CD-apigenin complex and 4′−OH in the *α*−CD complex are smaller than those in apigenin monomer. Consequently, the proton-donating abilities of 5−OH in the complex and 4′−OH in the *α*−CD complex are stronger than in apigenin alone. In contrast, opposite results are observed for 7−OH in the complex and 4′−OH in the *β*−CD complex.

### 3.5. SPLET Mechanism

#### 3.5.1. Proton Affinity

In the SPLET mechanism, the initial step involves the formation of the phenoxide anion through heterolytic bond dissociation of the hydroxyl group. PA is a crucial parameter representing the ability of proton donation. A lower PA value indicates a stronger proton-donating ability and, correspondingly, a higher antioxidative activity. The calculated PA values for various hydroxyl groups in apigenin across the studied phases are outlined in Table 4.

In apigenin monomer, the PA of 4′−OH is smaller than that of the other hydroxyl groups in the benzene phase. Conversely, in the other phases, 7−OH exhibits the lowest PA. These findings indicate that in the gas and benzene phases, proton abstraction from 4′−OH is easier than from the other hydroxyl groups in apigenin monomer. In polar phases, 7−OH is the most likely hydroxyl group to undergo the initial step of the SPLET mechanism.

In the *α*−CD-−apigenin complex, 4′−OH has a smaller PA than the other hydroxyl groups in the benzene, chloroform, and pyridine phases. Similarly, in the other phases, 7−OH exhibits the lowest PA. These results demonstrate that in the *α*−CD−apigenin complex, proton abstraction from 4′−OH is easier than from the other hydroxyl groups in the benzene, chloroform, and pyridine phases. In other phases, 7−OH is the most probable hydroxyl group to initiate the SPLET mechanism. In the *β*−CD−apigenin complex, 7−OH has a smaller PA than the other hydroxyl groups across all studied media. Consequently, in the *β*−CD−apigenin complex, 7−OH is most likely to undergo the initial step of the SPLET mechanism. These discussions reveal that the strongest antioxidative hydroxyl group of apigenin is influenced by the inclusion complexation with CD following the SPLET mechanism. Based on the lowest O−H PAs, the proton-donating ability sequence of apigenin is as follows: *β*−CD−apigenin > *α*−CD−apigenin > apigenin monomer. Therefore, *β*−CD−apigenin appears to be the most promising candidate to act as the scavenger following the SPLET mechanism, and the complexation with CD enhances the antioxidative ability of apigenin.

The PAs of 4′−OH, 5−OH, and 7−OH decrease after complexation with CD. Consequently, the CD complexation enhances the antioxidative ability of these hydroxyl groups in apigenin following the SPLET mechanism.

#### 3.5.2. Electron Transfer Enthalpy

In the SPLET mechanism, the second step involves electron transfer from the anion formed in the initial step, characterized by ETE. The calculated ETE values for various hydroxyl groups in apigenin are provided in Table 4.

In apigenin monomer, the ETE of 5−OH is smaller than that of the other hydroxyl groups in the benzene and chloroform phases. Conversely, in polar phases, 4′−OH exhibits the lowest ETE. These results indicate that in the benzene and chloroform phases, electron transfer from 5−OH is easier than from the other hydroxyl groups in the apigenin monomer anion, whereas 4′−OH is the most likely hydroxyl group to undergo the second step of the SPLET mechanism in polar phases. In the *α*−CD−apigenin complex, the ETE of 5−OH is smaller than that of the other hydroxyl groups in the studied phases, making 5−OH the most likely candidate to undergo the second step of the SPLET mechanism. In the *β*−CD−apigenin complex, the ETE of 4′−OH is smaller than that of the other hydroxyl groups across all studied media. Consequently, in the *β*−CD−apigenin complex, 4′−OH is most likely to undergo the second step of the SPLET mechanism. These discussions reveal that the strongest electron-donating group of apigenin anion is influenced by the inclusion complexation with CD following the SPLET mechanism. Based on the lowest O−H ETEs, the electron-donating ability sequence of apigenin anion is as follows: apigenin monomer > *α*−CD−apigenin > *β*−CD−apigenin. Therefore, apigenin monomer appears to be the most suitable candidate to undergo the second step of the SPLET mechanism, and the complexation with CD diminishes the electron-donating ability of apigenin anion.

The ETEs of 4′−OH, 5−OH, and 7−OH increase after complexation with CD. Consequently, the electron-donating abilities of these hydroxyl groups in the apigenin anion within the CD-apigenin complex are weaker than those in apigenin alone. The complexation with CD diminishes the electron-donating ability of the hydroxyl groups in apigenin anion.

## 4. Discussion

Ordinarily, free energy (∆_r_*G* = ∆_r_*H* − *T*∆_r_*S*) serves as the criterion for distinguishing thermodynamically favored processes. However, for the studied mechanisms, the absolute values of the entropic term (−*T*∆_r_*S*) only differ by a few units of kJ/mol, causing the free energies to be slightly shifted compared to the corresponding enthalpies (∆r*H*).

In multiple-step mechanisms, the first step holds thermodynamic significance. Enthalpies such as IP and PA are related to the initial step of SET−PT and SPLET mechanisms. By examining the lowest BDE, IP, and PA values presented in Table 2, Table 3 and Table 4, it can be concluded that the reaction enthalpies in the studied environments follow the following sequences:Chloroform phase: BDE < IP < PA;Benzene phase: BDE < PA < IP;Pyridine, ethanol, acetonitrile, DMSO and water phases: PA < BDE < IP.

The significantly lower BDE in apigenin compared to the corresponding IP and PA indicates the dominance of the HAT mechanism in the benzene and chloroform phases. In contrast, in the pyridine, ethanol, acetonitrile, DMSO, and water phases, the lowest PA of the hydroxyl groups in apigenin is significantly lower than the corresponding lowest BDE and IP. Hence, the SPLET mechanism emerges as the most thermodynamically probable reaction pathway in polar solvents.

Based on the findings from Section 3.3 and Section 3.5.1, it can be inferred that the sequence of the lowest BDE is as follows: apigenin monomer < *α*−CD−apigenin < *β*−CD−apigenin. Similarly, the sequence of the lowest PA is as follows: *β*−CD−apigenin < *α*−CD−apigenin < apigenin monomer. In the HAT and SPLET mechanisms, the antioxidative activities of flavonoids are determined by the lowest BDE and lowest PA, respectively. Therefore, in the benzene, and chloroform phases, the antioxidative activity of apigenin can be ranked in the following order: apigenin monomer > *α*−CD−apigenin > *β*−CD−apigenin. In polar phases, the order is reversed.

Based on the above discussion, it can be deduced that in the benzene and chloroform phases, the complexation with CD reduces the antioxidative activity of apigenin, whereas the opposite result occurs in polar phases. This conclusion is consistent with experimental findings indicating that CD inclusion complexation enhances the antioxidative activity of flavonoids in aqueous ethanol solutions [18,19].

We should also stress that experimentally observed behavior is not solely related to observed thermodynamics, but also the kinetics of radical scavenging play important roles [46], although the two aspects are usually in accordance.

## 5. Conclusions

This study aims to elucidate the differences in antioxidative activity between flavonoids and CD inclusion complexes using HAT, SET−PT, and SPLET mechanisms. Apigenin, a representative flavonoid, was selected for analysis employing DFT calculations. Prior to investigating various molecular descriptors, we explored the conformational properties of both apigenin and the CD−apigenin complex. Our analysis revealed the presence of closed-shell type intramolecular hydrogen bonds between apigenin and CD. The interaction between *β*−CD and apigenin was found to be stronger than that between *α*−CD and apigenin. Hydrogen bonds between *α*−CD and apigenin exhibited weak strength and an electrostatic dominant character, whereas those between *β*−CD and apigenin demonstrated medium strength and a dominant covalent interaction character. These observations contribute valuable insights into the antioxidative activity differences in flavonoids and their CD inclusion complexes.

Upon examining the optimized structures, it became evident that all hydroxyl groups in apigenin were susceptible to attack by free radicals. However, complexation with CD was found to diminish the hydrogen atom donating and electron transfer abilities while enhancing the proton donating ability of apigenin. The HAT mechanism predominates in the benzene and chloroform phases, whereas the SPLET mechanism emerges as the most thermodynamically probable reaction pathway in polar solvents. Consequently, in the benzene and chloroform phases, the complexation with CD diminishes the antioxidative activity of apigenin, whereas the opposite result occurs in polar phases.

This conclusion aligns with experimental findings demonstrating that CD inclusion complexation enhances the antioxidative activity of flavonoids in aqueous ethanol solutions.

## Figures and Tables

**Figure 1 antioxidants-12-02018-f001:**
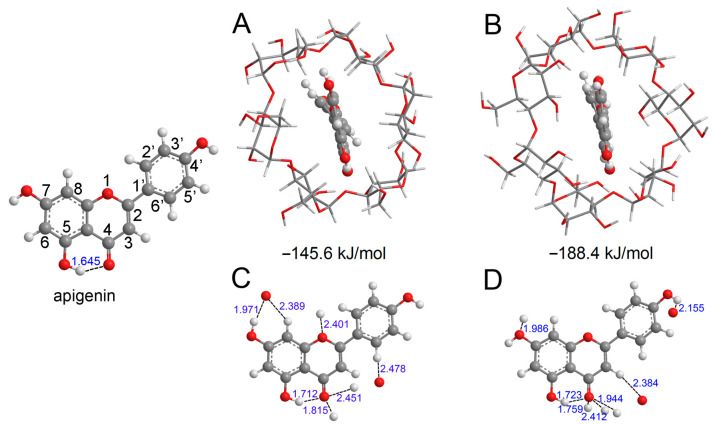
The optimized structure of apigenin and the most stable interaction geometries of the *α*-CD−apigenin complex (**A**,**B**) and *β*−CD−apigenin complex (**C**,**D**). To see more clearly about the interaction site, only apigenin and the interaction atom of CD were retained in (**C**,**D**). In these figures, the hydrogen, carbon and oxygen atoms are in the off-white, gray and red colors, respectively.

**Figure 2 antioxidants-12-02018-f002:**
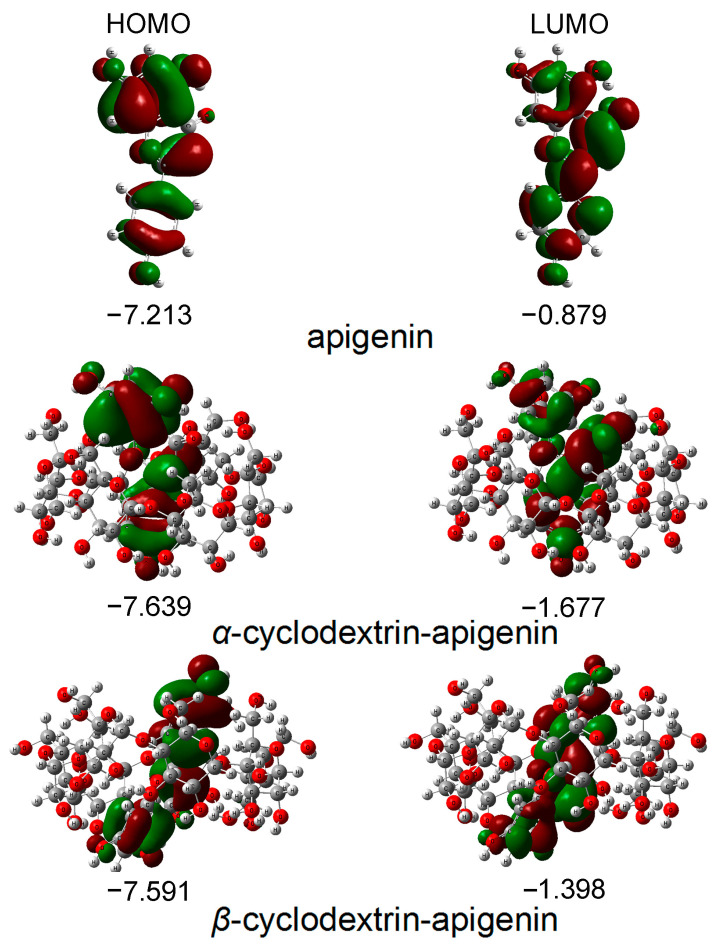
The energies and distributions of HOMO and LUMO orbitals for apigenin in the gas phase. In these figures, the hydrogen, carbon and oxygen atoms are in the off-white, gray and red colors, respectively.

**Table 1 antioxidants-12-02018-t001:** Some topological parameters in a.u at the BCPs of the hydrogen bonds.

Hydrogen Bond	*ρ* _BCP_	∇^2^*ρ*_BCP_	*V* _BCP_	*G* _BCP_	*H*_BCP_ (10^−3^)
apigenin					
O5−H5⋯O4	0.054	0.171	−0.050	0.046	−3.725
*α*−CD−apigenin					
O5−H5⋯O4	0.046	0.151	−0.043	0.040	−2.498
C1′−H1′⋯O	0.010	0.035	−0.007	0.008	0.704
O1⋯H−O	0.011	0.038	−0.008	0.009	0.474
O4⋯H−O	0.011	0.037	−0.008	0.009	0.558
O4⋯H−O	0.033	0.120	−0.029	0.029	0.790
O7−H7⋯O	0.024	0.085	−0.020	0.021	0.500
C8−H8⋯O	0.013	0.047	−0.010	0.011	0.794
*β*−CD−apigenin					
O5−H5⋯O4	0.046	0.148	−0.042	0.040	−2.617
C4′−H4′⋯O	0.017	0.058	−0.016	0.015	−0.730
C3−H3⋯O	0.012	0.036	−0.010	0.009	−0.250
O4⋯H−O	0.025	0.082	−0.023	0.022	−0.982
O4⋯H−O	0.012	0.036	−0.010	0.009	−0.270
O4⋯H−O	0.039	0.132	−0.036	0.034	−1.440
O7⋯H−O	0.024	0.076	−0.022	0.020	−1.291

**Table 2 antioxidants-12-02018-t002:** BDE in kJ/mol obtained by the M062X/6-311+G** method. The data in the form of the bold and underline represent the lowest BDEs.

	Benzene	Chloroform	Pyridine	Ethanol	Acetonitrile	DMSO	Water
apigenin							
BDE(4′−OH)	** 373.9 **	** 375.2 **	** 374.6 **	** 380.7 **	** 377.1 **	** 374.6 **	** 371.7 **
BDE(5−OH)	427.6	420.3	417.6	409.0	415.5	415.4	399.0
BDE(7−OH)	396.1	400.0	396.1	402.9	398.6	395.9	393.8
*α*-CD-apigenin							
BDE(4′−OH)	** 379.3 **	** 378.0 **	** 379.6 **	** 383.6 **	** 382.1 **	** 379.0 **	** 374.4 **
BDE(5−OH)	423.3	412.1	416.5	405.4	414.5	414.7	393.4
BDE(7−OH)	437.4	426.1	432.4	424.7	431.4	431.0	413.9
*β*-CD-apigenin							
BDE(4′−OH)	** 384.8 **	** 393.0 **	** 389.7 **	** 387.5 **	** 388.8 **	** 388.3 **	** 378.4 **
BDE(5−OH)	413.8	410.5	407.2	406.8	405.8	405.6	398.4
BDE(7−OH)	406.0	409.8	406.7	412.1	408.7	406.2	404.4

**Table 3 antioxidants-12-02018-t003:** IP and PDE in kJ/mol obtained by the M062X/6-311+G** method. The data in the form of the bold and underline represent the lowest PDEs.

	Benzene	Chloroform	Pyridine	Ethanol	Acetonitrile	DMSO	Water
apigenin							
IP	679.2	395.4	482.2	603.2	509.3	525.0	596.3
PDE(4′−OH)	** 118.2 **	** 225.4 **	** −92.7 **	** 6.5 **	** 35.1 **	**−58.7**	** 17.0 **
PDE(5−OH)	171.8	267.7	−49.7	38.0	72.4	−18.0	48.2
PDE(7−OH)	140.4	247.4	−71.3	28.7	56.6	−37.5	39.1
α−CD−apigenin							
IP	695.2	407.4	511.1	628.1	535.7	553.3	612.3
PDE(4′−OH)	** 108.9 **	** 212.3 **	**−116.6**	**−10.2**	** 13.6 **	**−82.6**	** 7.6 **
PDE(5−OH)	152.4	249.2	−79.7	15.2	47.0	−46.8	31.5
PDE(7−OH)	166.5	263.2	−63.7	31.0	62.9	−30.6	47.0
β−CD−apigenin							
IP	680.0	397.0	487.6	608.5	509.8	527.8	600.2
PDE(4′−OH)	** 139.8 **	** 242.0 **	**−83.0**	** 18.7 **	** 46.2 **	**−47.8**	** 27.5 **
PDE(5−OH)	158.8	259.5	−65.6	31.2	63.2	−30.5	38.7
PDE(7−OH)	151.1	258.8	−66.0	43.2	66.1	−29.8	53.5

**Table 4 antioxidants-12-02018-t004:** PA and ETE in kJ/mol obtained by the M062X/6-311+G** method. The data in the form of the bold and underline represent the lowest PAs and ETEs.

	Benzene	Chloroform	Pyridine	Ethanol	Acetonitrile	DMSO	Water
apigenin							
PA(4′−OH)	** 407.2 **	446.1	93.9	161.9	202.1	110.7	157.2
PA(5−OH)	472.1	482.7	129.3	180.6	231.7	141.9	176.2
PA(7−OH)	413.1	** 435.7 **	** 86.9 **	** 152.8 **	** 197.0 **	** 105.7 **	** 153.0 **
ETE(4′−OH)	391.0	183.6	** 301.5 **	** 459.9 **	** 348.5 **	** 361.4 **	** 464.3 **
ETE(5−OH)	** 379.7 **	** 182.0 **	303.1	462.6	351.0	364.5	467.6
ETE(7−OH)	407.3	208.7	329.4	495.1	379.1	391.4	500.3
*α*−CD−apigenin							
PA(4′−OH)	** 389.8 **	** 415.5 **	** 78.2 **	143.7	188.5	97.2	143.1
PA(5−OH)	443.3	468.6	126.5	167.9	228.3	141.6	160.5
PA(7−OH)	392.3	427.0	81.5	** 142.1 **	** 186.8 **	** 96.1 **	** 138.9 **
ETE(4′−OH)	424.8	204.1	316.3	474.2	360.8	373.5	478.4
ETE(5−OH)	** 404.2 **	** 188.0 **	** 304.9 **	** 468.8 **	** 353.4 **	** 365.4 **	** 469.8 **
ETE(7−OH)	469.4	243.5	360.4	506.2	401.7	417.0	506.3
*β*−CD−apigenin							
PA(4′−OH)	402.7	438.8	88.0	153.9	195.9	104.9	152.9
PA(5−OH)	388.6	434.2	82.8	158.3	191.7	100.2	156.4
PA(7−OH)	** 386.8 **	** 414.2 **	** 62.4 **	** 137.5 **	** 172.7 **	** 79.7 **	** 134.1 **
ETE(4′−OH)	** 416.3 **	** 191.3 **	** 310.7 **	** 471.1 **	** 354.0 **	** 369.3 **	** 476.7 **
ETE(5−OH)	449.4	220.7	339.2	482.9	381.3	397.1	488.1
ETE(7−OH)	463.5	240.0	359.2	508.9	403.3	418.3	515.8

## Data Availability

Data is contained within the article.

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
