# Peer review of "A DFT−Based Mechanism Analysis of the Cyclodextrin Inclusion on the Radical Scavenging Activity of Apigenin"

_antioxidants, 2023, doi:10.3390/antiox12112018_

Round 1

Reviewer 1 Report

Comments and Suggestions for Authors

Submitted work can be considered relevant for wide readership. However, there are some minor issues that should be considered.

Lines 63, 64: The sentence could be reformulated. It can evoke that more flavonoids have been investigated in the paper.

In Computational details, references for used approaches (B3LYP, M06-2X, SMD) and basis sets are missing.

Authors could mention dimensions of cavities of studied cyclodextrins.

Section 2.2 (eqs. 2, 3, …): Correct denotation of radical cation is radical-dot followed by + sign.

Lines 248, 249: Incorrect information. If the IP is too low, electron transfer to oxygen molecule is promoted and the substance shows pro-oxidant effect due to ROS formation.

Section 3.4.1: Authors should point out that in several solvents, IPs of apigenin and the beta-CD-apigenin are practically the same - the differences are lower than 4 kJ/mol and complexation may not affect electron-donating ability.

Section 3.4.2: Authors could mention that PDEs obey the trends found for BDEs.

In Discussion, authors should admit that experimentally observed behavior is not solely related to observed thermodynamics, but also the kinetics of radical scavenging plays important role, although the two aspects are usually in accordance. Here, authors could add some suitable references, e.g., Lucarini & Pedulli: Free radical intermediates in the inhibition of the autoxidation reaction. Chem. Soc. Rev. 39(6), 2106–2119 (2010). DOI:10.1039/b901838g or other works focused on both, thermodynamics and kinetics of antioxidant action.

Comments on the Quality of English Language

Some wording could be improved, see comments above.

Author Response

Comment 1: Lines 63, 64: The sentence could be reformulated. It can evoke that more flavonoids have been investigated in the paper.

Response: Thank you for your valuable suggestion. We have reformulated the corresponding sentence in the revised manuscript (page 2, line 65).

Comment 2: In Computational details, references for used approaches (B3LYP, M06-2X, SMD) and basis sets are missing.

Response: Thanks for your careful reading and helpful suggestions. We have added the related references (references 32, 33, 34 and 36) in the revised manuscript (page 2, lines 86, 88 and 94).

Comment 3: Authors could mention dimensions of cavities of studied cyclodextrins.

Response: Thanks a lot for your kind reminder. We have added the data of the cavities of studied cyclodextrins in the revised manuscript (page 2, lines 48 to 49).

Comment 4: Section 2.2 (eqs. 2, 3, …): Correct denotation of radical cation is radical-dot followed by + sign.

Response: Thank you for your careful reading and kind reminder. We have corrected the denotation of the radical cation in the revised manuscript (page 3, lines 111, 112, 116, 124 and 127).

Comment 5: Lines 248, 249: Incorrect information. If the IP is too low, electron transfer to oxygen molecule is promoted and the substance shows pro-oxidant effect due to ROS formation.

Response: We truly appreciate your professional comment. We have deleted the corresponding sentences in the revised manuscript (page 7, lines 249 to 250).

Comment 6: Section 3.4.1: Authors should point out that in several solvents, IPs of apigenin and the beta-CD-apigenin are practically the same - the differences are lower than 4 kJ/mol and complexation may not affect electron-donating ability.

Response: Thanks a lot for your professional guidance. We have added the information in the revised manuscript (page 8, lines 259 to 260).

Comment 7: Section 3.4.2: Authors could mention that PDEs obey the trends found for BDEs.

Response: Thank you for your valuable suggestion. We have mentioned that PDEs obey the trends found for BDEs in the revised manuscript (page 8, line 268).

Comment 8: In Discussion, authors should admit that experimentally observed behavior is not solely related to observed thermodynamics, but also the kinetics of radical scavenging plays important role, although the two aspects are usually in accordance. Here, authors could add some suitable references, e.g., Lucarini & Pedulli: Free radical intermediates in the inhibition of the autoxidation reaction. Chem. Soc. Rev. 39(6), 2106–2119 (2010). DOI:10.1039/b901838g or other works focused on both, thermodynamics and kinetics of antioxidant action.

Response: Thanks a lot for your kind reminder and the recommendation of the reported excellent work. We have added the information in the discussion section of the revised manuscript (page 10, lines 374 to 376) and the corresponding reference (reference 46) has also been cited.

Reviewer 2 Report

Comments and Suggestions for Authors

The authors report a camparative study on the antioidant ability of epigenin to exert the antioxidant ac tivity once include within CD.  The work is well planned and results well presented I have only ome suggestions:

- In the abstract please rephrase the sentence: "Initially, the study optimized the geometries of CD apigenin inclusion complexes, revealing the formation of hydrogen bonds which were meticulously analyzed."

- Figure 2: please added a rigth space between the FMO of each system.

- I believe that data obtained in gas phase can be removed from the paper, as they do not represent a real system. I found, instead, much more useful the employment of different solvent to explore the mechanisms of action

Comments on the Quality of English Language

With the exception of some sentences, the quality of english is satisfactory.

Author Response

Comment 1: In the abstract please rephrase the sentence: "Initially, the study optimized the geometries of CD apigenin inclusion complexes, revealing the formation of hydrogen bonds which were meticulously analyzed."

Response: Thanks for the reviewer’s careful reading. We have rephrased the corresponding sentence in the revised manuscript (page 1, lines 15 to 17).

Comment 2: Figure 2: please added a rigth space between the FMO of each system.

Response: We truly appreciate your professional comment. We have added a rigth space between the FMO of each system in Figure 2 in the revised manuscript (page 6, line 211).

Comment 3: I believe that data obtained in gas phase can be removed from the paper, as they do not represent a real system. I found, instead, much more useful the employment of different solvent to explore the mechanisms of action.

Response: Thanks a lot for your professional guidance. We have removed the data obtained in gas phase in the revised manuscript.